# *Toxoplasma gondii**ADSL* Knockout Provides Excellent Immune Protection against a Variety of Strains

**DOI:** 10.3390/vaccines8010016

**Published:** 2020-01-06

**Authors:** Luyao Wang, Ding Tang, Chenghang Yang, Jing Yang, Rui Fang

**Affiliations:** State Key Laboratory of Agricultural Microbiology, College of Veterinary Medicine, Huazhong Agricultural University, Wuhan 430070, Hubei, China; wly782@webmail.hzau.edu.cn (L.W.); tdtd@webmail.hzau.edu.cn (D.T.); SherlockYang@webmail.hzau.edu.cn (C.Y.); Yj0782@webmail.hzau.edu.cn (J.Y.)

**Keywords:** *Toxoplasma gondii*, adenylosuccinate lyase (*ADSL*), attenuated live vaccine, protective immunity

## Abstract

*Toxoplasma gondii* is a protozoan parasite, occurring worldwide, endangers human health and causes enormous economic losses to the Ministry of Agriculture. A safe and effective vaccination is needed to handle these problems. In addition, ideal vaccine production is a challenge in the future. In this study, we knocked out the adenylosuccinate lyase (*ADSL*) gene and found that the gene reduces the growth rate of *T. gondii* tachyzoites in vitro under standard growth conditions by plaque or replication experiments. Furthermore, mice that were immunized with tachyzoites of the ME49Δ*ADSL* strain induced 100% protection efficacy against challenge with the type 1 strain RH, type 2 strain ME49 and type 3 strain VEG. All mice that were immunized with ME49Δ*ADSL* had a survival rate of 100% when they were reinfected with wild-type strains, either 30 days or 70 days after immunization, and immunization was also protective against homologous infection with 50 *T. gondii* ME49 tissue cysts. In addition, the level of *Toxoplasma*-specific IgG was significantly elevated at 30 and 70 days after immunization. ME49Δ*ADSL* induced high levels of Th1 cytokines (interferon gamma (IFN-γ), interleukin (IL)-12) at 4 weeks after immunization and spleen cell cultures from mice vaccinated for 150 days were able to produce robust INF-γ and IL-12 levels in the supernatant. The results of the present study showed that Δ*ADSL* vaccination induced a T. *gondii*-specific cellular immune response against further infections. These results suggest that the ADSL-deficient vaccine can induce anti-*Toxoplasma gondii* humoral and cellular immune responses and has 100% immune protection against post-challenge by the type 1 strain RH, type 2 strain ME49 and type 3 strain VEG. It will be used as an excellent candidate for live vaccines and may contribute in a positive meaning to control human toxoplasmosis.

## 1. Introduction

*Toxoplasma gondii* is an obligate intracellular parasite that is distributed throughout the world and infects almost all warm-blooded animals [1]. The majority of *T. gondii* infection is asymptomatic in hosts with normal immune function, but systemic infections are more common in immunodeficient patients, such as those with AIDS, organ transplants, malignant tumors, etc. [2,3,4]. Congenital toxoplasmosis can also cause miscarriage, premature birth, teratogenic effects or stillbirth in pregnant women, especially in early pregnancy, and the incidence of teratogenesis is high for fetuses [5]. In animals, millions of lambs are still lost worldwide due to the miscarriage of ewe caused by toxoplasmosis, and huge economic losses are caused in agriculture [6,7]. Ethylamine, sulfonamide and other drugs can inhibit *T. gondii* folic acid metabolism but are ineffective for cysts, which will develop resistance over time [8]. Therefore, the development of a *T. gondii* vaccine is urgently needed for the prevention and control of toxoplasmosis and the reduction of economic losses. Currently, the live tachyzoites of strain S48 is a commercially available vaccine (Toxovax^®^), which has been used to reduce neonatal mortality in lambs [9,10]. Although the exact mechanism of the Toxovax^®^ vaccine is not completely clear, the success of Toxovax^®^ is a milestone in the development of a *Toxoplasma* vaccine. In the past ten years, research on *T. gondii* vaccines, such as protein, DNA vaccine and recombinant vaccines, has been carried out. Some soluble or secretory proteins are obtained from cultured tachyzoites as killed vaccines or native parasite antigens. However, these vaccines did not provide sufficient protection [11,12]. Then the recombinant DNA technology is an alternative strategy. A large number of *Toxoplasma* recombinant antigens have been widely tried. These include dense-granule proteins (GRAs), micronemes (MICs) and surface antigens (SAGs) [13,14,15]. Although these subunit vaccines were easier to produce and more stable than native antigens vaccines, they only induced partial immune protection against further parasite infection [16,17]. Thus, vaccine of safe and effective is urgently needed for *T. gondii*. In recent years, genetically modified strains as vaccines have become a major research focus. Many scientists have begun to study genetically modified parasites, and the virulence of *T. gondii* was weakened by deleting certain genes, as has been achieved in some studies [18,19,20,21,22]. In addition, early studies have shown that *T. gondii* is a purine auxotroph organism [23,24]. Hence, *T. gondii* possesses significant machinery for purine salvage, and this extensive machinery may enable this parasite to survive and replicate in an extensive range of mammalian cell types [25]. Adenylosuccinate lyase (*ADSL*) is present in the *T. gondii* salvage pathway to make IMP from AMP by a two-step reaction [25].

In this study, the *ADSL* gene was knocked out, and the protective immunity of the ME49Δ*ADSL* strain was comprehensively evaluated to obtain an excellent vaccine to confront *T. gondii* infection in mice models .The results showed that the *ADSL* gene reduces the growth rate of *T. gondii* tachyzoites in vitro and induces anti-*Toxoplasma* humoral and cellular immune responses, and the strain induces a 100% efficient protective immune response against multiple *Toxoplasma gondii* strains, including the virulent strain RH, whether in the context of long-term or short-term. The modified strain will be used as an excellent potential toxoplasmosis vaccine that is able to protect animals from further infections of a variety of *T. gondii* strains.

## 2. Materials and Methods

### 2.1. Mice and Parasite Strains

Eight-week-old female ICR mice were obtained from the Huazhong Agricultural University Laboratory Animal Center, HuBei, China. All animals were housed in specific pathogen-free conditions. Mice were adapted to the environment for one week before use. All animal experiments were permitted by the ethics committee of Huazhong Agricultural University (HZAUMO-2019-009). The *Toxoplasma gondii* type I strain RHΔ*hxgprt*, type 2 strain ME49, type 3 strain VEG, RFLP genotype *ToxoDB* #9 strain C7719, *ToxoDB* #3 strain TgPIG-WH1 and mutant strain ME49*:ΔADSL* used in this study were maintained in vitro in human foreskin fibroblast (HFF) cells (purchased from the ATCC, Rockefeller, MD, USA) and were propagated in Dulbecco’s modified Eagle’s medium (DMEM; Life Technologies, Inc., Rockville, MD, USA) supplemented with 10% fetal bovine serum (FBS), penicillin and streptomycin. All strains and cells were culture using standard techniques [26].

### 2.2. Disruption of ADSL by the CRISPR-Cas9 System

*ADSL* (TGME49_277760) deletions in ME49 strains were performed by the CRISPR/Cas9 system, which utilized homologous recombination to replace the target gene, and the associated method was performed as described previously [27,28]. Briefly, the resistance cassette loxp-DHFR*-loxp was amplified from pDONR-G265-loxP-DHFR*-loxp, and 1-kbp 5′- and 3′-homology arms were obtained from the genomic DNA from ME49 strains. These fragments were concatenated into the pUC19 vector and then assembled into the p*ADSL*:loxp-DHFR*-loxp plasmid, which can serve as a template to replace ADSL with loxp-DHFR*-loxp. All the primers used during plasmid construction are listed in Table 1, and all plasmids were proven to work by DNA sequencing before use. Furthermore, we modified pSAG1:CAS9-U6:sgUPRT to replace the UPRT targeting guide RNA (gRNA) with a single-guide RNA (sgRNA) that disrupted the *ADSL* gene by using Q5 site-directed mutagenesis. *Toxoplasma gondii* ME49 was transfected with the linearized p*ADSL*:loxp-DHFR*-loxp homologous fragment and ADSL-specific CRISPR plasmid as previously described [27,28] and selected with 1 µM pyrimethamine (Life Technologies, Inc., Rockville, MD, USA).

### 2.3. Virulence Tests and Cyst Formation of Mutants Versus WT Strains in Mice

Freshly egressed tachyzoites of the ME49 strain and ME49Δ*ADSL* strain were filtered with a 3.0 μm membrane filter and then harvested to intraperitoneally (ip) infect 8-week-old female ICR mice (10 mice per strain). Subsequently, the clinical symptoms of toxoplasmosis were observed, and mortality was recorded for 30 days. Blood samples were collected from the mouse tail vein at 30 days. After clot retraction, serum samples were collected and detected by enzyme-linked immunosorbent assay (ELISA) with *T. gondii* soluble antigens as coating antigen to confirm infection. Soluble *Toxoplasma* antigens were prepared by a previously described method [29]. Subsequently, brain cysts from seropositive mice were harvested, and DBA-FITC (Vector Laboratories, Burlingame, CA, USA) dyeing as previously described [30] was used to calculate the number of *Toxoplasma* cysts. The results of brain cyst number and cumulative mortality determination in the control and experimental group mice were graphed as Kaplan–Meier survival plots and then analyzed in Prism 5 (GraphPad Software Inc., La Jolla, CA, USA).

### 2.4. Toxoplasma Plaque Assay

HFF monolayer cells were seeded into 6-well plates and cultured at 37 °C with 5% CO_2_. Purified tachyzoites were quantified with a hemocytometer under a Nikon Eclipse TS100 phase contrast microscope (Nikon Instruments, Tokyo, Japan) and then inoculated onto host cells (200 parasites per well). Subsequently, the six-well plates containing parasites were placed at 37 °C with 5% CO_2_ for 13 days. After incubation, HFF cells were fixed, stained with 0.5% crystal violet solution, washed with PBS and imaged.

### 2.5. Assay for Parasite Replication In Vitro

The growth ability of wild-type andME49△ADSL strains was detected as previously described [31]. Briefly, tachyzoites that were grown in HFF cells for 3 days were purified by 3.0 μm membrane filtration and then inoculated in a 24-well plate that was seeded with HFF cells. After samples were incubated for 1 h at 37 °C with 5% CO_2,_ the parasites that failed to invade cells were washed away, and the cultures were continued for 24 h at 37 °C. Subsequently, rabbit anti-*TgALD* and mouse anti-*SAG1* were used as primary antibodies to detect extracellular parasites and total parasites, respectively. *SAG1* has been reported to be anchored to the parasite membrane [32,33]. Alexa 488-conjugated goat anti-mouse and Alexa 594-conjugated goat anti-rabbit IgGs were used to stain as a secondary antibodies by diluting 1:1000. The number of parasites per vacuole was calculated under fluorescence microscopy. A minimum of 100 total parasitophorous vacuoles was examined for one sample. All strains were tested three times independently.

### 2.6. Protection of the ME49ΔADSL Strain Against Acute Infection, Toxoplasma-Specific lgG levels and Cytokine Detection

In this experiment, the immune protection of ME49Δ*ADSL* tachyzoites against acute *T. gondii* infection was studied. Briefly, female ICR mice were immunized with 100 me49Δ*ADSL* tachyzoites intraperitoneally. Then, 32 and 72 days after immunization, seropositive mice were challenged intraperitoneally with 500 tachyzoites of RHΔ*hxgprt* or ME49 or 1 × 10^4^ tachyzoites of C7719, VEG or TgPIG-WH1 or orally with 50 fresh cysts of ME49, as previously described [22]; natural mice that were not immunized were used as controls. In addition, serum samples were collected from Δ*ADSL*-immunized mice 30 days and 70 days after immunization. Sera from mice that were not immunized were used as a control, and then these sera were used to assay the level of secreted interleukin-12p70 (IL-12p70), interferon gamma (INF-γ) and IL-4 by commercial ELISA kits following the manufacturer’s instructions (BioLegend, Inc., USA). *T. gondii-*specific IgG was measured by enzyme-linked immunosorbent assay (ELISA) as described previously [29].

### 2.7. Cytokine Production by Splenocytes After T. gondii Antigen Stimulation

One hundred fifty days after ICR mouse immunization, the spleens of seropositive mice were collected and ground gently in RPMI 1640 medium (Life Technologies, Inc., Rockville, MD, USA) with a 45 µm filter. Subsequently, the samples were centrifuged for 10 min at 1000× *g* and then resuspended in 10 mL of red cell lysis solution (Biosharp, Inc., Beijing, China) for 5 min at ambient temperature. The splenocyte suspensions were centrifuged as above and washed with RPMI 1640 medium. Trypan blue was used to assay the viable cells, and 3 × 10^5^ viable splenocytes that were counted by a hemocytometer under a Nikon Eclipse TS100 phase contrast microscope (Nikon Instruments, Tokyo, Japan) were seeded into 96-well flat-bottom tissue culture plates in a final volume of 100 µL of RPMI 1640 supplemented with 20% FBS (Life Technologies, Inc., Rockville, MD, USA) and 1% penicillin–streptomycin mixture. After 1 day of incubation, TSA (final concentration of 50 µg/mL) was used to stimulate the splenocytes. Meanwhile, concanavalin A (final concentration of 5 µg/mL) and 20% FBS RPMI 1640 medium were added to stimulate the splenocytes as positive and negative controls, respectively. All supernatants were collected after stimulation to detect the level of cytokines, which was performed as above. In addition, non-immunized naïve mice were subjected to the same experiment as a control.

### 2.8. Statistical Analysis

All data were analyzed in Prism 5 (GraphPad Software, Inc., La Jolla, CA, USA) using Student’s *t* tests, Gehan–Breslow–Wilcoxon tests or one-way ANOVA with Bonferroni post-tests, as indicated in figure legends.

## 3. Results

### 3.1. Generation of ADSL-Deficient Type II T. gondii

To study the biological function of the *ADSL* gene against *Toxoplasma gondii*, we generated a *Tg*ADSL mutant by using the CRISPR/Cas9-mediated genome editing system Figure 1a. A CRISPR plasmid containing the *ADSL* gene sgRNA and homology template *ADSL*:DHFR* was simultaneously transferred into the type II strain ME49. Subsequently, pyrimethamine was used to select stable single clones, which were confirmed successfully by diagnostic polymerase chain reactions (PCRs) Figure 1b. The correct integration of the DHFR marker at the *ADSL* locus was demonstrated by PCR1 and PCR2. The validation of PCR3 further proved the successful acquisition of the *ADSL* mutant strain Figure 1b. To measure the impact of *Tg**ADSL* disruption on parasite growth, we performed plaque assays. Compared with the parental strain, the knockout strain showed obvious growth defects according to plaques and replication results Figure 1c–e, suggesting that the *ADSL* gene plays an important role in the in vitro growth of *Toxoplasma gondii*.

### 3.2. Effects of ADSL Mutation on Virulence and Reduced Cyst Formation In Vivo

The consequences of ADSL deletion on the acute infection stage of the parasite were studied by performing a survival curve analysis. Briefly, 100 tachyzoites of both the parental strain ME49 and *ADSL* deletion mutants that were infected intraperitoneally into mice, and the survival rate results are shown in Figure 2a. That the survival rates at 30 days for the ME49 and Δ*ADSL* mutants groups were 20% and 70% Figure 2a, respectively, indicated that ADSL deletion led to attenuated virulence of the parasite. Subsequently, brains were collected in survivor mice at 30 days post-infection, and brain cysts were quantified by DBA-FITC staining. The cyst number in Δ*ADSL* mutants was significantly reduced and was compared with that of the ME49 strain under the same experimental conditions. The result is shown in Figure 2b.

### 3.3. Immune Responses of Immunized Mice to Acute Tachyzoites Infection

The results suggest that the virulence of the *ADSL* mutant was weakened, as described above. To examine whether mice after *ADSL* immunization were resistant to acute infection of *Toxoplasma gondii*, ICR mice were challenged intraperitoneally with 500 tachyzoites of RHΔ*hxgprt*, 1 × 10^3^ tachyzoites of ME49, or 1 × 10^4^ tachyzoites of VEG 30 days after immunization with the *ADSL* mutants. Meanwhile, naïve mice were challenged with the same strain and the same corresponding dose as a control. The mortality rates for non-immunized mice challenged with RH were 100%. The mortality rates of ME49- and VEG-challenged naïve mice were 80% and 20%, respectively. By contrast, the survival rate of all immunized mice was 100%, with infection with any of the three strains, and no obvious clinical symptoms of toxoplasmosis were observed during 40 days of infection Figure 3a–c.

Subsequently, the long-time protective immunity was studied for immunized ICR mice. Briefly, ICR mice at 72 days after immunization with ME49Δ*ADSL* were challenged with 500 tachyzoites of RHΔ*hxgprt* or ME49 or 1 × 10^4^ tachyzoites of VEG. The survival of the mice was monitored for another 40 days. The survival of all immunized mice was 100% for the three strains, and no symptoms were observed post-infection. However, the mortality rate of naïve mice infected with the RH and type II ME49 strains was 100%, while VEG caused 30% mortality Figure 3d–f.

The above results not only indicate that short- and long-term protection were furnished by ME49Δ*ADSL* vaccination against acute infection but also suggest that the Δ*ADSL* strain provided protection against bradyzoite infection.

### 3.4. T. gondii-Specific IgG Antibodies and Cytokine Production After Vaccination

To further explore the immunogenicity of Δ*ADSL*, the specific anti-*T. gondii* IgG antibody level was detected by ELISA in sera samples that were acquired from vaccinated mice at 30 and 70 days. The results showed that the IgG titer in immunized mice was significantly higher than that in non-immunized mice at 30 and 70 days Figure 4. However, IgG levels were relatively stable over time (Figure 4) for immunized mice. The results suggested that Δ*ADSL* strain infection elicited a strong humoral response. Subsequently, we also collected vaccinated mouse sera at 30 and 70 days to measure the levels of Th1 (IFN-γ and IL-12) and Th2 (IL-4) cytokines by ELISA. The IFN-γ-IL-12 axis, serving as a representative of Th1-type cells, mediates the main cellular immune response to *T. gondii*. The levels of these cytokines were significantly elevated at 30 and 70 days compared to those in control mice (Figure 5a–c). Mice immunized for 70 days showed a significant decrease in IFN-γ levels compared to those immunized for 30 days (Figure 5a). In summary, these results suggested that vaccination with Δ*ADSL* induced cellular and humoral immune responses against secondary infections of *T. gondii*.

### 3.5. Cytokine Production After Vaccination in Response to Stimulation with T. gondii Antigen

To verify the antigen recall response of immunized mice, we checked the cytokine levels of splenocytes that were infected with the *ADSL* mutant for 150 days and non-immunized mice that were used as a control. The splenocytes were stimulated with TSA prepared as described above, and culture supernatants were used to detect the IFN-γ and IL-12 levels by ELISA. The results showed that TSA stimulated high levels of the pro-inflammatory cytokines INF-γ and IL-12 (Figure 6a,b), while non-stimulated or nonvaccinated splenocytes served as a control. IFN-γ is a major regulator of cell-mediated immunity that activates hematopoietic and nonhematopoietic effector cells to control parasite replication, as previously reported [34,35,36]. In our study, we found that INF-γ and IL-12 were produced from vaccinated splenocytes upon TSA stimulation. During a second infection, effective cellular immunity is activated and makes a major contribution.

## 4. Discussion

It is well known that the purine salvage pathways are essential for the growth and development of *T. gondii* [23,24,25,37]. *T. gondii* possesses many enzymes involved in interconversion and salvage of host purines, and adenylosuccinate lyase (*ADSL*) is one of them [25]. However, the role of the *ADSL* gene in the growth of *T. gondii* is unclear. In this study, *ADSL* was knocked out by using the gene editing system CRISPR/Cas9 in the ME49 strain. Subsequently, the results of plaque and replication assays showed that the gene reduces the growth rate of *T.*
*gondii* tachyzoites in vitro under standard growth conditions. In addition, virulence experiments indicate that the replication of the *ADSL* mutant strain in vivo was slowed down and the ability to form cysts was weakened. The specific mechanism for these results is unclear, and we suspect that the reduction of AMP synthesis required for *T.*
*gondii* due to the *ADSL* deficiency [23] thereby reduces the precursors of nucleotide synthesis and affects the growth of *Toxoplasma gondii*. Next, the Δ*ADSL* mutant was investigated for potential immunization as a live-attenuated vaccine against reinfection of the *T.*
*gondii* wild-type strain in a mouse model. Δ*ADSL*-infected mice survived when the dose of infection was 100. Therefore, for all subsequent experiments, 100 ME49Δ*ADSL* tachyzoites were used as the vaccination dose, which not only provided a balance between safety and high immunogenicity but also represented a promising attenuated live vaccine strain.

We then tested the antibody and cytokine levels induced in Δ*ADSL*-immunized mice by ELISA. These results suggested a combination of humoral and cellular immune responses and their protective effects against *T. gondii* attack, but the cellular immune response was dominant. Δ*ADSL*-immunized mice evoked a high level of anti-*T. gondii* IgG at 30 and 70 days after immunization, indicating that IgG antibodies play an important protective role in further infections with *T. gondii* tachyzoites. Initiation of IL-12 and IFN-γ production in host immune cells is essential for host resistance to parasites [38]. In our study, we found that Δ*ADSL*-immunized mice produced higher levels of IL-12, IFN-γ, and IL-4 at 30 and 70 days post-infection than non-immunized mice. It is well known that the production of IFN-γ and IL-12 that is due to infection by *T. gondii* stimulates T cells, macrophages, dendritic cells, and neutrophils [38,39]. In addition, the Th2 cytokine IL-4 is critical for neutralizing the inflammation and immunopathologies caused by the Th1 response [40]. The balance of Th1 and Th2 cytokine levels was observed to control tachyzoite transmission in *ADSL*-immunized mice. *ADSL*-immunized mice produced immune protection against post-challenge of other various *Toxoplasma gondii* strains, such as the type I strain RHΔ*hxgprt*, type 2 strain ME49 and type 3 strain VEG (i.e., 100% survival rate), whereas nonvaccinated mice died within 10 dpi. It is noteworthy that this immune protection can provide complete long-term or short-term protection against the virulent strain RH Δ*hxgprt*. We detected cytokines from spleen cell culture supernatants at day 150 post-Δ*ADSL* immunization and found a significant increase in the level of the IFN-γ and IL-12 cytokines after stimulation with TSA. The results showed that immunized mice could activate cellular immunity and clear secondary infections efficiently better than non-immunized mice. In addition, we found that Δ*ADSL* immunization induced high levels of IL-12 and IFN-γ for a fairly long duration after infection, and this effect may be caused by the unique propagation dynamics of the Δ*ADSL* mutants in vivo. Therefore, the continued IL-12 and IFN-γ induction might be a consequence of these parasite activities, and prolonged IL-4 may be to provide balance.

In conclusion, we demonstrated that a single ip vaccination of 100 ME49:Δ*ADSL* tachyzoites provided immune protection against challenge with various *T. gondii* strains by stimulating cellular and humoral immune responses. In addition, ME49:Δ*ADSL* immunization could improve the survival rate and reduce cyst production compared with those of unimmunized groups. Moreover, the Δ*ADSL* mutant also generated high levels of Th1 (IFN-γ and IL-12) immunity and elevated Th2 (IL-4) protective immunity, as indicated at 30 and 70 days post-immunization, and protected animals from further infections with various *T. gondii* wild-type strains. Although our research from mice models demonstrated the Δ*ADSL* mutant vaccine could be used as an excellent potential vaccine, further investigations are needed to check the safety of vaccination and efficiency of protection in other host species like swine and sheep.

## Figures and Tables

**Figure 1 vaccines-08-00016-f001:**
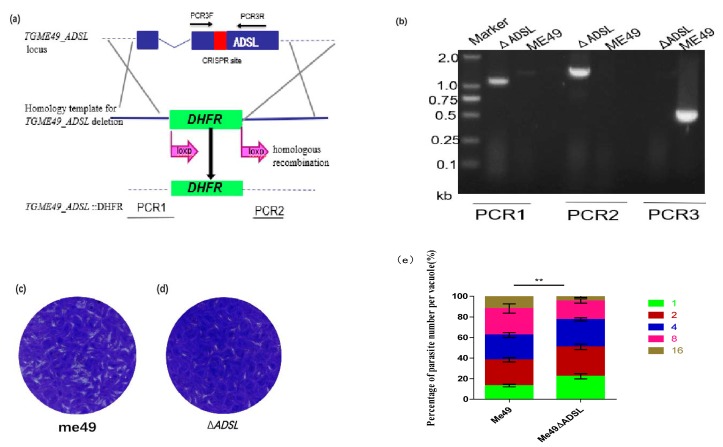
Generation and analysis of an *ADSL* deletion mutant. (**a**) Schematic illustration of knocking out *ADSL* in ME49 to produce Δ*ADSL*. (**b**) Diagnostic polymerase chain reaction (PCR) for a Δ*ADSL* mutant clone. (**c**, **d**) Plaque assay comparing the growth of the Δ*ADSL* mutant to that of the wild-type strain ME49. (**e**) Tachyzoite replication under standard tissue culture conditions. HFF cells were infected with purified tachyzoites, invading parasites were cultured for 24 h, and then fluorescently stained to determine the number of parasites in each parasitophorous vacuole (PV), ** *p* < 0.01, two-way analysis of variance.

**Figure 2 vaccines-08-00016-f002:**
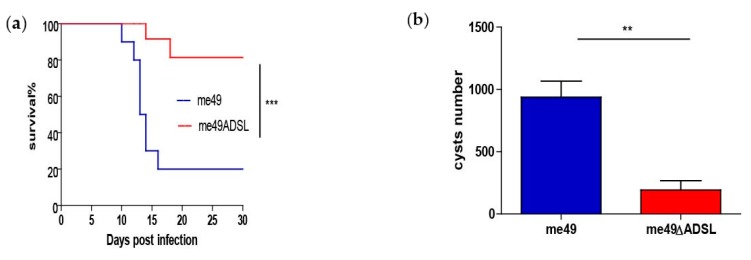
Virulence and cyst formation of the Δ*ADSL* mutant in mice. (**a**) Virulence assessment of the indicated strains in ICR mice, as shown in Figure 2a. (**b**) Brain cyst counts for the mice in () that survived at day 30, as shown in Figure 2b. ** *p* < 0.01, *** *p* < 0.001, Gehan–Breslow–Wilcoxon tests.

**Figure 3 vaccines-08-00016-f003:**
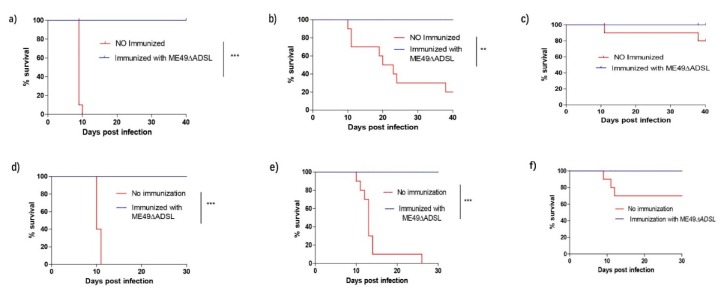
Δ*ADSL* parasite immunization protected mice from *Toxoplasma gondii* tachyzoite infection. ICR mice were pre-immunized with 100 tachyzoites of the ME49 Δ*ADSL* mutant. (**a**–**c**) Thirty days post-immunization, the immunized mice were challenged with 500 RH tachyzoites (**a**), 10^3^ ME49 tachyzoites (**b**) or 10^3^ VEG tachyzoites ((**c**); 10 mice for each strain) by intraperitoneal injection, and their survival was monitored for another 40 days. (**d**–**f**) 75 days post-immunization, 500 tachyzoites of the RH Δ*hxgprt* (**d**), ME49 (**e**) or VEG (**f**) strains were used to challenge the mice, and mouse survival was monitored for another 30 days. Non-immunized mice were included as controls. ** *p* < 0.01 *** *p* < 0.001, Gehan–Breslow–Wilcoxon tests.

**Figure 4 vaccines-08-00016-f004:**
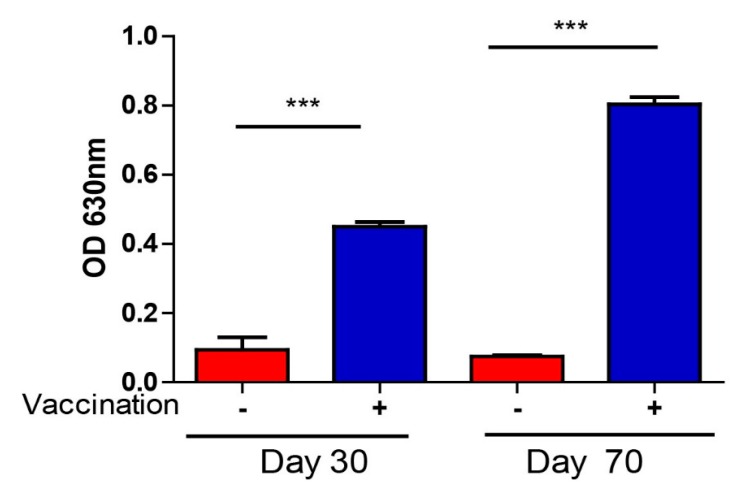
Relative levels of *Toxoplasma*-specific IgG, as determined by indirect ELISA. Sera from naïve mice were used as controls, and three mice from each group were analyzed. **** p* < 0.001, NS: not significant, Student’s *t*-test.

**Figure 5 vaccines-08-00016-f005:**
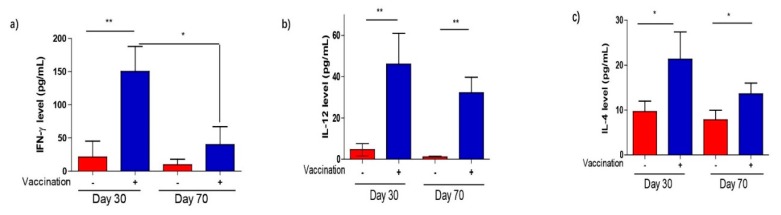
Levels of selected cytokines in the sera of mice 30 and 70 days after Δ*ADSL* immunization. (**a**–**c**) Levels of the indicated cytokines measured by ELISA. Sera from naïve mice were used as controls, and three mice from each group were analyzed. * *p* < 0.05, ** *p* < 0.01, NS: not significant, Student’s *t*-test.

**Figure 6 vaccines-08-00016-f006:**
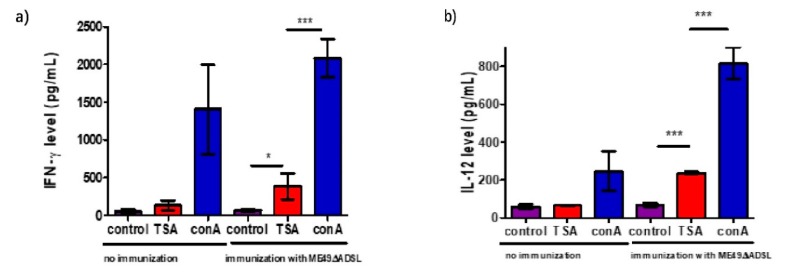
Cytokine production by splenocytes of ME49 Δ*ADSL*-vaccinated mice after *Toxoplasma* antigen stimulation. ICR mice were vaccinated with 100 tachyzoites of ME49 Δ*ADSL*, and splenocytes were harvested 150 days post-vaccination. The in vitro cultured splenocytes were then treated with fetal bovine serum (control), total soluble *Toxoplasma* antigen (TSA) or concanavalin A (conA, positive control) for 72 h. Subsequently, the levels of IFN-γ (**a**) and IL-12 (**b**) in the culture supernatants were measured by ELISA. Splenocytes isolated from non-immunized naïve mice were included as controls. Three mice from each group were analyzed, and each treatment condition was repeated three times. * *p* < 0.05, *** *p* < 0.001, Student’s *t*-test.

**Table 1 vaccines-08-00016-t001:** Primers used in this study.

Primer	Sequence	Use
gRNA-ADSL-Fw	5′- GACACTGCTCAATTTCGTCGGTTTTAGAGCTAGAAATAGC-3′	To construct the ADSL specific CRISPR plasmid
gRNA-R	5′-AACTTGACAT CCCCA TTTAC-3′	To construct the *ADSL* specific CRISPR plasmid
pUC19-Fw pUC19-Rv	5′-GGCGTAATCATGGTCATAGC-3′ 5′-CTCGAATTCACTGGCCGTCG-3′	Amplification of pUC19 for Gibson assembly Amplification of pUC19 for Gibson assembly
loxp-DHFR*-loxp-Fw loxp-DHFR*-loxp-Rv 5′-GGACACGCTGAACTTGTGGC-3′	5′-CAACCCGCGCAGAAGACATC-3′	Amplification of loxp-DHFR*-loxp for Gibson assembly Amplification of loxp-DHFR*-loxp for Gibson assembly
U5ADSL-Fw pADSL: loxp- *DHFR**-loxp construction	5′-GTTGTAAAACGACGGCCAGTCTAATTTTTGCCGGGTCTGG-3′	Amplification of 5′-homology of *ADSL* for
U5ADSL-Rv pADSL: loxp- *DHFR**-loxp construction	5′-GATGTCTTCTGCGCGGGTTG CGAGACGAAGAAGAGAGC-3′	Amplification of 5′-homology of *ADSL* for
U3ADSL-Fw pADSL: loxp -*DHFR**-loxp construction	5′-GCCACAAGTTCAGCGTGTCCACCTCAGTTGTCGGCACCGT-3′	Amplification of 3′-homology of *ADSL* for
U3ADSL-Rv pADSL: loxp--*DHFR**-loxp construction	5′-GCTATGACCATGATTACGCCCGCAGACTCAAATCTATTC-3′	Amplification of 3′-homology of *ADSL* for
5′-UpU5 ADSL	5′-GCTCTTGCTTTCGTCGCTGTC-3′	PCR1 of Δ*ADSL:DHFR*
3′-In*DHFR**-Fw	5′-CGTGACCACGCCAAAGTAG-3′	PCR1 of Δ*ADSL:DHFR*
5′-In*DHFR**-Rv	5′-GCACTTGCAGGATGAATTCC-3′	PCR2 of Δ*ADSL:DHFR*
3′-DnU3ADSL	5′-GTGTCTCGCACATGCGCGTT-3′	PCR2 of Δ*ADSL:DHFR*
5′-UpgRNA ADSL	5′- AACGCAATCAAGACGCTGGC-3′	PCR3 of Δ*ADSL:DHFR*
3′-DngRNA ADSL	5′- CGTCACTGTACCAGCGAGCAG-3′	PCR3 of Δ*ADSL:DHFR*

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
