# Peer review of "Toxoplasma gondii**ADSL* Knockout Provides Excellent Immune Protection against a Variety of Strains"

_vaccines, 2020, doi:10.3390/vaccines8010016_

Round 1
Reviewer 1 Report
The paper describes the efficacy of a vaccine obtained knocking out ADSL of Toxoplasma gondii. The topic is of interest but the manuscript requires a deep revision.
Line 10 - "is a protozoan parasite, occurring worldwide
line 34 - please rewrite the sentence in a more proper manner. T. gondii is not usually intracellularly parasitic parasite (what does it mean?)
line 35 - Infection instead of toxoplasmosis (toxoplasmosis is a disease and a disease cannot be asymptomatic), please, modify
line 42 - T. gondii
line 45 - Cysts are parasite stages occurring during chronic phase, when the pathogenetic potential of the parasite is reduced, so the set up of a vaccine would be not recommended for this stage, but for tachyzoites, responsible for the spread of infection throughout the host's tissues. Please, modify the sentence.
line 47 - a vaccine is not an agent, please modify
line 49 - the vaccines can only solve part of the problem. Please clarify, the readers cannot be aware of the weakness of this vaccine, so the importance of a further vaccine is not underlined enough.
lines 60-61 - what is the toxoplasma problem? please, write in a more correct way
line 66 - to solve toxoplasmosis is not correct, please modify the sentence
line 108 - Toxoplasma or toxoplasma
Author Response
Dear reviewer,
We have carefully studied your valuable comments and have done our best to revise the manuscript.
Please refer to the attachments. There are three attachments. They are a point-by-point response to the reviewer’s comments, Manuscripts with revision marks, and the new revised manuscript(no mark).
Special thanks to you for your good comments

Reviewer 2 Report
This study investigated a genetically modified T. gondii strain with CRISPR/CAS gene editing. The deletion of ADSL-gene resulted in strong humoral and cellular immune response in infected/vaccinated mice. The study design seems to be sufficient enough to conclude, that vaccination with modified live vaccine is protective enough to prevent infections with field strains.
I suggest to exchange "humoural" with humoral through out the whole manuscript, since it is hardly not used anymore.
In general I suggest to rewrite the conclusions in the abstract and in the conclusion section (Line 28-29, Line 306-314) as it is to my opinion to early to conclude that it is a perfect vaccine. You do not have any results from other host species, such as swine and sheep and different other mammals. The long term effect cannot be predicted from a 40 day period. Despite it seems promising, I would rather conclude just for mice and talk of further research need in other host species
The rest of my comments and remarks can be found directly in the pdf-document.

Author Response
Response to Reviewer 2 Comments
Dear reviewer,
We have carefully studied your valuable comments and have done our best to revise the manuscript. The point to point responds to the reviewer’s comments are listed as following:
All the modifications are highlighted in the new manuscript.
Point1:I suggest to exchange "humoural" with humoral through out the whole manuscript, since it is hardly not used anymore
Response1: Thank you very much. According to your comment, we have had the manuscript corrected the mistakes.
Point 2: In general I suggest to rewrite the conclusions in the abstract and in the conclusion section (Line 28-29, Line 306-314) as it is to my opinion to early to conclude that it is a perfect vaccine. You do not have any results from other host species, such as swine and sheep and different other mammals. The long term effect cannot be predicted from a 40 day period. Despite it seems promising, I would rather conclude just for mice and talk of further research need in other host species
Response2: Thank you for your careful work. According to comment, we have carefully corrected this sentence in the abstract.
Moreover, we strongly agree with you that our research is currently performed in mouse models and further research is needed in other host species. We have carefully rewritten this conclusion.
In fact, during our experiments, the mice immunized with the ΔADSL strain survived for about 150 days.
Point 3:Please change the last sentence as it sounds like advertisement. It destroys to my opinion the scientific topic of your study. I am absolutely with you that your vaccine has potential, but would not expect that it is the solution of toxoplasmosis. Rather talk about that your vaccine candidate may contribute in a positive sence to control human toxoplasmosis.
Response3: Thank you for your careful work. According to comment, we have carefully corrected this sentence.
Point4: I don’t understand what you mean with "infectious toxoplasmosis". what is the difference between toxoplasmosis and infectious toxoplasmosis???
Response4:Thank you for your valuable and thoughtful comments.We are sorry for this language mistake. We have corrected this sentence in the new manuscript.
Point5: two times "in addition" sounds not professional. Please rewrite!
strain 、form 、Me49 、immunity、
“into” delete
Response5: Thank you very much for your suggestion, we have modified these words
Special thanks to you for your good comments

Round 2
Reviewer 1 Report
the reviewer's suggestions have been followed, so the manuscript can be accepted